# Semantic Exploration from Language Abstractions and Pretrained Representations

**Allison C. Tam**
DeepMind
London, UK
actam@deepmind.com

**Neil C. Rabinowitz**
DeepMind
London, UK
ncr@deepmind.com

**Andrew K. Lampinen**
DeepMind
London, UK
lampinen@deepmind.com

**Nicholas A. Roy**
DeepMind
London, UK
nroy@deepmind.com

**Stephanie C. Y. Chan**
DeepMind
London, UK
scychan@deepmind.com

**DJ Strouse**
DeepMind
London, UK
strouse@deepmind.com

**Jane X. Wang**[*]
DeepMind
London, UK
wangjane@deepmind.com

**Andrea Banino**[*]
DeepMind
London, UK
abanino@deepmind.com

**Felix Hill**[*]
DeepMind
London, UK
felixhill@deepmind.com

## Abstract

Effective exploration is a challenge in reinforcement learning (RL). Novelty-based exploration methods can suffer in high-dimensional state spaces, such as continuous partially-observable 3D environments. We address this challenge by defining novelty using semantically meaningful state abstractions, which can be found in learned representations shaped by natural language. In particular, we evaluate vision-language representations, pretrained on natural image captioning datasets. We show that these pretrained representations drive meaningful, task-relevant exploration and improve performance on 3D simulated environments. We also characterize why and how language provides useful abstractions for exploration by considering the impacts of using representations from a pretrained model, a language oracle, and several ablations. We demonstrate the benefits of our approach with on- and off-policy RL algorithms and in two very different task domains— one that stresses the identification and manipulation of everyday objects, and one that requires navigational exploration in an expansive world. Our results suggest that using language-shaped representations could improve exploration for various algorithms and agents in challenging environments.

## 1 Introduction

Exploration is one of the central challenges of reinforcement learning (RL). A popular way to increase an agent's tendency to explore is to augment trajectories with intrinsic rewards for reaching novel environment states. However, the success of this approach depends critically on which states are considered novel, which can in turn depend on how environment states are represented.

The literature on novelty-driven exploration describes several approaches to deriving state representations [7]. One popular method employs random features and represents the state by embedding

---

[*]Equal contribution

36th Conference on Neural Information Processing Systems (NeurIPS 2022).

the visual observation with a fixed, randomly initialized target network [Random Network Distillation; 6]. Another method uses learned visual features, taken from an inverse dynamics model [Never Give Up; 3]. These approaches work well in classic 2D environments like Atari, but it is less clear whether they are as effective in high-dimensional, partially-observable settings such as 3D environments. For instance, in 3D settings, different viewpoints of the same scene may map to distinct visual states/features, despite being semantically similar. The difficulty of identifying a good mapping between visual state and feature space is exacerbated by the fact that useful state abstractions are highly task dependent. For example, a task involving tool use requires object-affordance abstractions, whereas navigation does not. Thus, acquiring state representations that support effective exploration is a chicken-and-egg problem—knowing whether two states should be considered similar requires the type of understanding that an agent can only acquire after effectively exploring its environment.

To overcome these challenges, we propose giving agents access to prior knowledge during training, in the form of abstractions derived from large vision-language models [e.g. 41] that are pretrained on image captioning data. We use these pretrained models to derive a intrinsic reward that reflects meaningful novelty. We hypothesize that representations acquired by vision-language pretraining drive effective, semantic exploration in 3D environments, because the representations are shaped by the unique abstract nature of natural language.

Several aspects of natural language suggest that it could be useful to direct novelty-based exploration. First, language is inherently abstract: language links superficially distinct, but causally-related situations by describing them similarly, and contrasts between causally-distinct states by describing them differently, thus outlining useful concepts [29, 28]. Second, humans use language to communicate important information efficiently, without overspecifying [20, 21]. Thus, human language omits distracting irrelevant information and focuses on important aspects of the world. For example, it is often observed that an agent rewarded for seeking novel experience would be attracted forever to a TV with uncontrollable and unpredictable random static [7]. However, a human would likely caption this scene "a TV with no signal" regardless of the particular pattern; thus an agent exploring with language abstractions would quickly leave the TV behind. Figure 1 shows another conceptual example of how language abstractions can accelerate exploration.

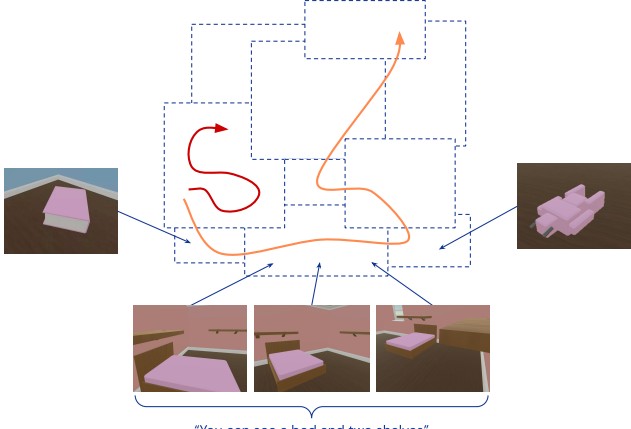

Figure 1: Navy dashed lines delineate semantically meaningful states. By using representations that align well with these boundaries (i.e. language), then agents more effectively explore the wider state space (orange trajectory). If the representations do not reflect these boundaries and instead are amenable to visual noise (i.e. different colors, viewpoints, etc.), then agents may only focus on a visually novel, yet narrow subset of states (red trajectory).

We first perform motivating proof-of-concept experiments using a language oracle. We show that language is a useful abstraction for exploration not only because it coarsens the state space, but also because it coarsens the state space in a way that reflects the semantics of the environment. We then demonstrate that our results scale to environments without a language oracle using pretrained vision encoders, which are only supervised with language during pretraining. This work strives to enhance the representations used in novelty-based exploration, rather than compare various exploration methods.

We consider two popular novelty-based exploration methods from the literature, Never Give Up (NGU; Badia et al. [3]) and Random Network Distillation (RND; Burda et al. [7]), and compare them to their language-augmented variants, Lang-NGU/LSE-NGU and Lang-RND. We evaluate performance and sample efficiency on object manipulation, search, and navigation tasks in two challenging 3D environments simulated in Unity: Playroom (a house containing toys and furniture) and City (a large-scale urban setting). Our results show that language-based exploration with pretrained vision-language representations improves sample efficiency on Playroom tasks by 18-70%. It also doubles

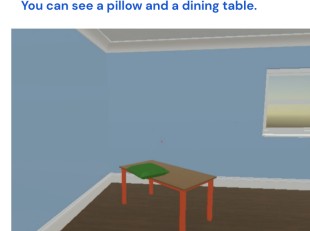 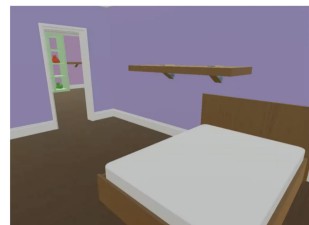 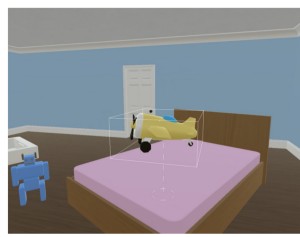

You can see a pillow and a dining table. You can see a bed, two shelves, a book case, a teddy, and a rubber duck. You are holding a plane. You can see a bed, a robot, and a storage tray.

(a) Example instances of $O_V$ and $O_L$ from the Playroom environment.

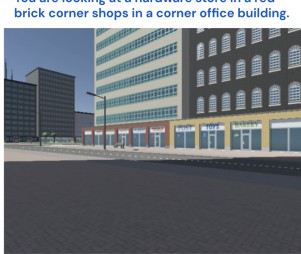 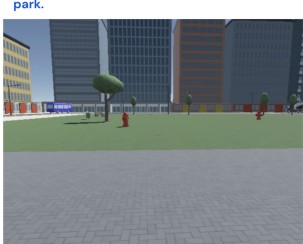 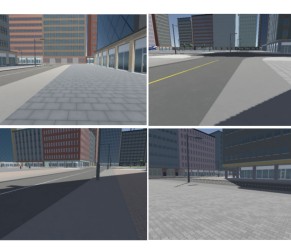

You are looking at a hardware store in a red brick corner shops in a corner office building. You are looking at a green grass ground in a park. You are looking at a road.

(b) Example instances of $O_V$ and $O_L$ from City. Many different scenes can be associated with the same caption.

Figure 2: Visual observations from the environment and example captions generated by the language oracle. Appendix Figure S4 contains more example captions.

the visited areas in City, compared to baseline methods. We show that language-based exploration is effective for both on-policy (IMPALA [17]) and off-policy (R2D2 [25]) agents.

## 2 Related Work

**Exploration in RL**   Classical exploration strategies include $\epsilon$-greedy action selection [51], state-counting [49, 4, 32, 30, 3], curiosity driven exploration [44], and intrinsic motivation methods [36]. Our work is part of this last class of methods, where the agent is given an intrinsic reward for visiting diverse states over time [35]. Intrinsic rewards can be derived from various measures: novelty [43, 55, 6, 56], prediction-error [38, 3], ensemble disagreement [11, 39, 48, 46, 18, 50], or information gain [23]. One family of methods gives intrinsic reward for following a curriculum of goals [8, 12, 40]. Others use novelty measures to identify interesting states from which they can perform additional learning [16, 54]. These methods encourage exploration in different ways, but they all rely on visual state representations that are learned jointly with the policy. Although we focus on novelty-based intrinsic reward and demonstrate the benefits of language in NGU and RND, our methodology is relatively agnostic to the exploration method. We suggest that many other exploration methods could be improved by using language abstractions and pretrained embeddings to represent the state space.

**Pretraining representations for RL**   Pretraining has been used in RL to improve the representations of the policy network. Self-supervised representation learning techniques distill knowledge from external datasets to produce downstream features that are helpful in virtual environments [15, 53]. Some recent work shows benefits from pretraining on more general, large-scale datasets. Pretrained CLIP features have been used in a number of recent robotics papers to speed up control and navigation tasks. These features can condition the policy network [26], or can be fused throughout the visual encoder to integrate semantic information about the environment [37]. The goal of these works is to improve perception in the policy. Pretrained language models can also provide useful initializations for training policies to imitate offline trajectories [42, 27]. These successes demonstrate that large pretrained models contain prior knowledge that can be useful for RL. While the existing literature uses pretrained embeddings directly in the agent, we instead allow the policy network to learn from scratchm and only utilize pretrained embeddings to guide exploration during training (Figure S2). We imagine that future work may benefit from combining both approaches.

**Language for exploration**   Some recent works have used language to guide agent learning, by either using language subgoals for exploration/planning or providing task-specific reward shaping [47, 33, 13, 19]. Schwartz et al. [45] use a custom semantic parser for VizDoom and show that representing states with language, rather than vision, leads to faster learning by simplifying policy inputs. Chaplot et al. [10] tackle navigation in 3D by constructing a semantic map of the environment from pretrained SLAM modules, language-defined object categories, and agent location. This approach lends itself to navigation, but it is unclear how it would extend easily to more generic settings or other types of tasks, such as manipulation. Work concurrent to ours by Mu et al. [34] shows how language, in the form of hand-crafted BabyAI annotations, can help improve exploration in 2D environments. These works demonstrate the value of language abstractions: the ability to ignore extraneous noise and highlight important environment features. However, these prior methods rely on environment-specific semantic parsers or annotations, which may limit the settings to which they can be applied. In contrast, by exploiting powerful pretrained vision-language models, our approach can be applied to *any* visually-naturalistic environment, including 3D settings, which have not been widely studied in prior exploration work. We additionally do not require any language from the environment itself. Our method could even potentially improve exploration for physical robots, but we leave that for future work.

## 3   Method

We consider a goal-conditioned Markov decision process defined by a tuple $(\mathcal{S}, \mathcal{A}, \mathcal{G}, P, R_e, \gamma)$, where $\mathcal{S}$ is the state space, $\mathcal{A}$ is the action space, $\mathcal{G}$ is the goal space, $P : \mathcal{S} \times \mathcal{A} \to \mathcal{S}$ specifies the environment dynamics, $R_e : \mathcal{S} \times \mathcal{G} \to R_e$ is the extrinsic reward, and $\gamma$ is the discount factor. State $\mathbf{s_t}$ is presented to the agent as a visual observation $O_V$. In some cases, in order to calculate intrinsic reward, we use a language oracle $\mathcal{O} : \mathcal{S} \to \mathcal{L}$ that provides natural language descriptions of the state, $O_L$. Note that $O_L$ is distinct from the language instruction $g \in \mathcal{G}$, which is sampled from a goal distribution at the start of an episode—the agent never observes $O_L$. We later remove the need for a language oracle by using pretrained models.

We use goal-conditioned reinforcement learning to produce a policy $\pi_g(\cdot \mid O_V)$ that maximizes the expected reward $\mathbb{E}[\sum_{t=0}^{H} \gamma^t (r_t^e + \beta r_t^i)]$, where $H$ is the horizon, $r_t^e$ is the extrinsic reward, $r_t^i$ is the intrinsic reward, and $\beta$ is a tuned hyperparameter. The intrinsic reward is goal-agnostic and is computed with access to either $O_V$ or $O_L$. Note that neither $O_L$ nor pretrained embeddings are used by the policy, and thus we only use them during training to compute the intrinsic reward (Figure 3).

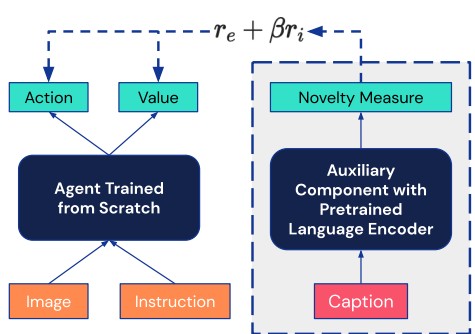

Our approach builds on two popular exploration algorithms: Never Give Up (NGU; Badia et al. [3]) and Random Network Distillation (RND; Burda et al. [7]). These algorithms were chosen to demonstrate the value of language under two different exploration paradigms. While both methods reward visiting novel states, they differ on several dimensions: the novelty horizon (episodic versus lifetime), how the history of past visited states is retained (non-parametric versus parametric), and how states are represented (learned controllable states versus random features).

Figure 3: The agent is trained from scratch using RL to optimize extrinsic and intrinsic reward. It acts using the image observation $O_V$ and goal $g$. During training, the novelty-based intrinsic reward is calculated using an auxiliary component that does not share parameters with the agent (dashed box). The auxiliary component may incorporate a pretrained language (pictured above) or image encoder, which may respectively require $O_L$ or $O_V$. The latter does not rely on language provided by the environment. See Figure S2 for more details.

### 3.1   Never Give Up (NGU)

To more clearly isolate our impact, we focus only on the episodic novelty component of the NGU agent [3]. State representations along the trajectory are written to a non-parametric episodic memory

buffer. The intrinsic reward reflects how novel the current state is relative to the states visited so far in the episode. Novelty is a function of the L2 distances between the current state and the $k$-nearest neighbor representations stored in the memory buffer. Intrinsic reward is higher for larger distances.

Full details can be found in the original paper; however, we make two key simplifications. While Badia et al. [3] proposes learning a family of policy networks that are capable of different levels of exploration, we train one policy network that maximizes reward $r = r_e + \beta r_i$ for a fixed hyperparameter $\beta$. We also fix the long-term novelty modulator $\alpha$ to be 1, essentially removing it.

The published baseline method, which we refer to as **Vis-NGU**, uses a controllable state taken from an inverse dynamics model. The inverse dynamics model is trained jointly with the policy, but the two networks do not share any parameters.

Table 1: Summary of NGU variants.

| Name | Embedding Type | Required Input |
|---|---|---|
| Vis-NGU | Controllable State | Vision |
| Lang-NGU | BERT | Language |
| | $CLIP_{text}$ | Language |
| | $ALM_{text}$ | Language |
| LSE-NGU | $CLIP_{image}$ | Vision |
| | $ALM_{image}$ | Vision |

Table 2: Summary of family of RND-inspired methods. Intrinsic reward is derived from the prediction error between the trainable network and frozen target function.

| Name | Trainable Network | Target Function |
|---|---|---|
| Vis-RND | $f_V : O_V \to \mathbb{R}^k$ | randomly initialized, fixed $\hat{f}$ |
| ND | $f_{\{V,L\}} : O_{\{V,L\}} \to \mathbb{R}^k$ | pretrained $ALM_{\{image, text\}}$ |
| Lang-RND | $f_L : O_L \to \mathbb{R}^k$ | randomly initialized, fixed $\hat{f}$ |
| LD | $f_C : O_V \to O_L$ | $O_L$ from language oracle |

The intrinsic reward relies on directly comparing state representations from the buffer, so our approach focuses on modifying the embedding function to influence exploration (Table 1). **Lang-NGU** uses a frozen pretrained language encoder to embed the oracle caption $O_L$. We compare language embeddings from BERT [14], CLIP [41], Small-ALM, and Med-ALM. The ALMs (ALign Models) are trained with a contrastive loss on the ALIGN dataset [24]. Small-ALM uses a 26M parameter ResNet-50 image encoder [22]; Med-ALM uses a 71M parameter NFNet [5]. The language backbones are based on BERT and are all in the range of 70-90M parameters. We do not finetune on environment-specific data; this preserves the real world knowledge acquired during pretraining and demonstrates its benefit without requiring any environment-specific captions.

**LSE-NGU** does not use the language oracle. Instead, it uses a frozen pretrained image encoder to embed the visual observation $O_V$. We use the image encoder from CLIP or ALM, which are trained on captioning datasets to produce outputs that are close to the corresponding language embeddings. The human-generated captions structure the visual embedding space to reflect features most pertinent to humans and human language [31], so the resulting representations can be thought of as Language Supervised Embeddings (LSE). The primary benefit of LSE-NGU is that it can be applied to environments without a language oracle or annotations. CLIP and ALM are trained on real-world data, so they would work best on realistic 3D environments. However, we imagine that in future work the pretraining process or dataset could be tailored to maximize transfer to a desired target environment.

## 3.2 Random Network Distillation (RND)

Our RND-inspired family of methods rewards lifetime novelty. Generically, the intrinsic reward is derived from the prediction error between a trainable network and some target value generated by a frozen function (Table 2). The trainable network is learned jointly with the policy network, although they do not share any parameters. As the agent trains over the course of its lifetime, the prediction error for frequently-visited states decreases, and the associated intrinsic reward consequently diminishes. Intuitively, the weights of the trainable network implicitly store the state visitation counts.

For clarity, we refer to the baseline published by Burda et al. [7] as **Vis-RND**. The trainable network $f_V : O_V \to \mathbb{R}^k$ maps the visual state to random features. The random features are produced by a fixed, randomly initialized network $\hat{f}_V$. Both $f_V$ and $\hat{f}_V$ share the same architecture: a ResNet followed by a MLP. The intrinsic reward is the mean squared error $\|f_V(O_V) - \hat{f}_V(O_V)\|^2$.

In network distillation (**ND**), the target function is not random, but is instead a pretrained text or image encoder from CLIP/ALM. The trainable network $f$ learns to reproduce the pretrained representations. To manage inference time, $f$ is a simpler network than the target (see Appendix A.2). The intrinsic loss is the mean squared error between $f$ and the large pretrained network. Like the respective Lang-NGU and LSE-NGU counterparts, text-based ND requires a language oracle, but image-based ND does not.

In Section 5.1 we compare against two additional methods to motivate why language is a useful abstraction. The first, **Lang-RND**, is a variant in which the trainable network $f_L : O_L \rightarrow \mathbb{R}^k$ maps the oracle caption to random features. The intrinsic reward is the mean squared error between the outputs of $f_L$ and fixed $\hat{f}_L$ with random initialization. Both $f_L$ and $\hat{f}_L$ networks are of the same architecture.

The second method, language distillation (**LD**), is loosely inspired by RND in that the novelty signal comes from a prediction error. However, instead of learning to produce random features, the trainable network learns to caption the visual state, i.e. $f_C : O_V \rightarrow O_L$. The network architecture consists of a CNN encoder and LSTM decoder. The intrinsic reward is the negative log-likelihood of the oracle caption under the trainable model $f_C$. In LD, the exploration dynamics not only depend on how frequently states are visited but also the alignment between language and the visual world. We test whether this caption-denoted alignment is necessary for directing semantic exploration by comparing LD to a variant with shuffled image-language alignment (**S-LD**) in Section 5.1.

# 4 Experimental Setup

## 4.1 Environments

Previous exploration work benchmarked algorithms on video games, such as 2D grid-world MiniHack and Montezuma's Revenge, or 3D first-person shooter Vizdoom. In this paper, we focus on first-person Unity-based 3D environments that are meant to mimic familiar scenes from the real world (Figure 2).

**Playroom**    Our first domain, Playroom [1, 52], is a randomly-generated house containing everyday household items (e.g. bed, bathtub, tables, chairs, toys). The agent's action set consists of 46 discrete actions that involve locomotion primitives and object manipulation, such as holding and rotating.

We study two settings in Playroom. In the first setting, the agent is confined to a single room with 3-5 objects and is given a `lift` or `put` instruction. At the start of an episode, the set of objects are sampled from a larger set of everyday objects (i.e. a candle, cup, hairdryer). Object colors and sizes are also randomized, adding superficial variance to different semantic categories. The instructions take the form: "Lift a <object>" or "Put a <object> on a {bed, tray}". With a `lift` goal, the episode ends with reward 1 or 0 whenever any object is lifted. With a `put` goal, the episode ends with reward 1 when the condition is fulfilled. This setting tests spatial rearrangement skills.

In the second setting, the agent is placed in a house with 3-5 different rooms, and is given a `find` instruction of the form "Find a {teddy bear, rubber duck}". Every episode, the house is randomly generated with the teddy and duck hidden amongst many objects, furniture, and decorations. The target objects can appear in any room— either on the floor, on top of tables, or inside bookshelves. The agent is randomly initialized and can travel throughout the house and freely rearrange objects. The episode ends with reward 1 when the agent pauses in front of the desired object. The `find` task requires navigation/search skills and tests the ability to ignore the numerous distractor objects.

**City**    Our second domain, City, is an expansive, large-scale urban environment. Each episode, a new map is generated; shops, parks, and buildings are randomly arranged in city blocks. Daylight is simulated, such that the episode starts during the morning and ends at nighttime. The agent is randomly initialized and is instructed to "explore the city." It is trained to maximize its intrinsic reward and can take the following actions: `move_{forward,backward,left,right}`, `look_{left,right}`, and `move_forward_and_look_{left,right}`. We divide up the map into a $32 \times 32$ grid and track how many unique bins are visited in an episode.

Additionally, City does not provide explicit visual or verbal signage to disambiguate locations. As such, systematic exploration is needed to maximize coverage. In contrast to Playroom, City tests long horizon exploration. A Playroom episode lasts only 600 timesteps, whereas a City episode lasts 11,250 and requires hundreds of timesteps to fully traverse the map even once. The City covers a 270-by-270 meter square area, which models a 2-by-2 grid of real world blocks.

## 4.2 Captioning Engine

We equip the environment with a language oracle that generates language descriptions of the scene, $O_L$, based on the Unity state, $s$ (Figure 2). In Playroom, the caption describes if and how the agent

interacts with objects and lists what is currently visible to it. In City, $O_L$ generally describes the object that the agent is directly looking at, but the captions alone do not disambiguate the agent's locations. Since these captions are generated from a Unity state, these descriptions may not be as varied or rich as a human's, but they can be generated accurately and reliably, and at scale.

### 4.3 Training Details

At test time, the agent receives image observation $O_V$ and language-specified goal $g$. The policy network never requires caption $O_L$ to act. During training, the exploration method calculates the intrinsic reward from $O_L$ or $O_V$.

We show that language-based exploration is compatible with both policy gradient and Q-learning algorithms. We use Impala [17] on Playroom and R2D2 on City [25]. Q-learning is more suitable for the City, because the action space is more restricted compared to the one needed for Playroom tasks.

For both environments, the agent architecture consists of an image ResNet encoder and a language LSTM encoder that feed into a memory LSTM module. The policy and value heads are MLPs that receive the memory state as input. If the exploration method requires additional networks, such as the trainable network in RND or inverse dynamics model in NGU, they do not share any parameters with the policy or value networks. Figure S2 is a visualization of an Impala agent that uses language-augmented exploration. Hyperparameters and additional details are found in Appendix A.

## 5 Results

### 5.1 Motivation: Language is a Meaningful Abstraction

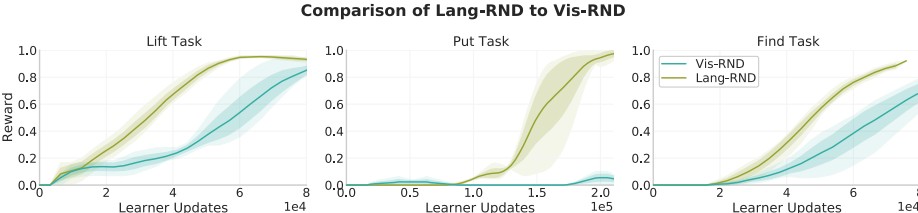

(a) Lang-RND outperforms Vis-RND by creating a coarser, more compact state space.

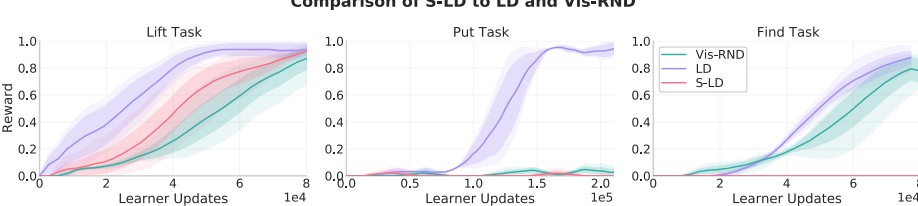

(b) LD outperforms S-LD. It is important how language abstractions carve up the state space.

Figure 4: Our comparisons demonstrate that language is useful for exploration, because it outlines a more abstract, semantically-meaningful state space. Results are shown with a 95% confidence band.

We share the intuition with other work [e.g. 34] that language can improve exploration. We design a set of experiments to show how and why this may be the case. Our analysis follows the desiderata outlined by Burda et al. [6]—prediction-error exploration ought to use a feature space that filters irrelevant information (compact) and contains necessary information (sufficient). Burda et al. [6] specifically studies RND and notes that the random feature space, the outputs of the random network, may not fully satisfy either condition. As such, we use the language variants of RND to frame this discussion.

We hypothesize that language abstractions are useful, because they (1) create a coarser state space and (2) divide the state space in a way that meaningfully aligns with the world (i.e. using semantics). First, if language provides a coarser state space, then the random feature space becomes more compact,

leading to better exploration. We compare Lang-RND to Vis-RND to test this claim. Lang-RND learns the `lift` task 33% faster and solves the `put` task as Vis-RND starts to learn (Figure 4a).

Second, we ask whether semantics – that is *how* language divides up the state space – is critical for effective exploration. We use LD to test this hypothesis, precisely because the exploration in LD is motivated by modeling the semantic relationship between language and vision.

We compare LD to a shuffled variant S-LD, where we replace the particular semantic state abstraction that language offers with a statistically-matched randomized abstraction (Figure 5). S-LD is similar to LD; the intrinsic reward is the prediction error of the captioning network. However, instead of targeting the language oracle output, the S-LD trainable network produces a different target caption $\widetilde{O_L}$ that may not match the image. $\widetilde{O_L}$ is produced by a fixed, random mapping $\hat{f}_S : O_V \to \widetilde{O_L}$. $\hat{f}_S$ is constrained such that the marginal distributions $P(O_L) \approx P(\widetilde{O_L})$ are matched under trajectories produced by policy $\pi_{LD}$. See Appendix A.4 for full details on the construction of S-LD.

Thus, whereas the LD captions parcel up state space in a way that reflects the abstractions that language offers, the randomized mapping $\hat{f}_S$

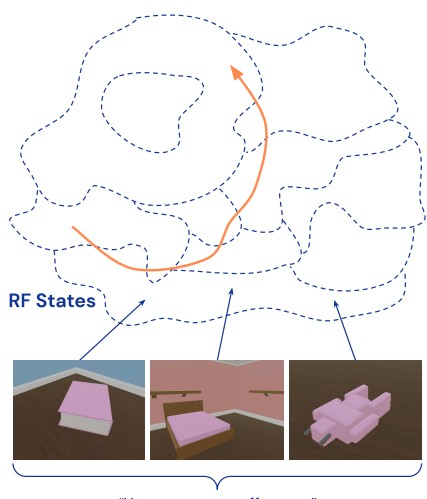

"You can see a coffee cup"

Figure 5: The dotted lines correspond to state abstractions given by the shuffled $\hat{f}_S$ used in S-LD. The states are grouped together based on similarities in the visual random feature space and assigned a label. Exploring in this shuffled space is less effective than exploring with the semantically-meaningful abstractions shown in Figure 1.

parcels up state space in a way that abstracts over random features of the visual space (Figure 5). We control for the compactness and coarseness of the resulting representation by maintaining the same marginal distribution of captions.

If semantics is crucial for exploration, then we expect to see LD outperform S-LD. This indeed holds experimentally (Figure 4b). We can also view these results under the Burda et al. [6] framework. The S-LD abstractions group together visually similar, but semantically distinct states. A single sampled caption likely fails to capture the group in a manner that is representative of all the encompassing states. In other words, $\hat{f}_S$ produces a compact feature space that may not be sufficient. This may explain why S-LD learns faster than Vis-RND on the simpler `lift` task but fails on the more complex `put` and `find` tasks. The S-LD experiments imply that language abstractions are helpful for exploration because they expose not only a more compact, but also a more semantically meaningful state space.

## 5.2 Pretrained Vision-Language Representations Improve Exploration

Having shown how language can be helpful for exploration, we now incorporate pretrained vision-language representations into NGU and RND to improve exploration. Such representations (e.g. from the image encoder in CLIP/ALM) offer the benefits of explicit language abstractions, without the need to rely on a language oracle. We also compare language-shaped representations to pretrained ImageNet embeddings to isolate the effect of language. To keep the number of experiments tractable, we only perform a full comparison on the Playroom tasks.

**City**   We first compare how representations affect performance in a pure exploration setting. With no extrinsic reward, the agent is motivated solely by the NGU intrinsic reward to explore the City. We report how many unique areas the agent visits in an episode in Figure 6. While optimizing coverage only requires knowledge of an agent's global location rather than generic scene understanding, vision-language representations are still useful simply because meaningful exploration is inherently semantic. Lang-NGU, which uses text embeddings of $O_L$, visits an area up to 3 times larger. LSE-NGU achieves 2 times the coverage even without querying a language oracle (Appendix Figure S5).

**Playroom** We next show that pretrained vision-language representations significantly speed up learning across all Playroom tasks (Figure 7). The LSE-NGU and Lang-NGU agents improve sample efficiency by 50-70% on the `lift` and `put` tasks and 18-38% on the `find` task, depending on the pretraining model used. The ND agents are significantly faster than Vis-RND, learning 41% faster on the `find` task. We also measure agent-object interactions. Nearly all LSE-NGU and Lang-NGU agents learn to foveate on and hold objects within 40k learning updates, whereas Vis-NGU agent takes at least 60k updates to do so with the same frequency (Appendix Figure S7). Although LSE-NGU and image-based ND agents do not access a language oracle, they are similarly effective as their annotation-dependent counterparts in the Playroom tasks (Appendix Figure S6), suggesting that our method could be robust to the availability of a language oracle.

To demonstrate the value of rich language, we compare LSE-NGU agents to a control agent that instead uses pretrained ImageNet embeddings from a 70M NFNet [5]. ImageNet embeddings optimize for single-object classification, so they confer some benefit to the most object-focused tasks, `lift` and `put`. However, Ima-

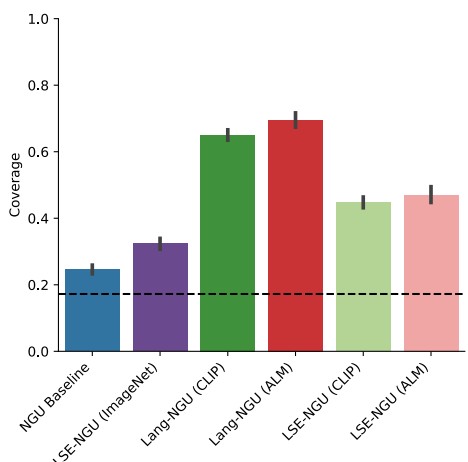

Figure 6: Coverage of City (number of bins reached on map) by NGU variants using different state representations for exploration, normalized by coverage of a ground-truth agent. The ground-truth agent represents state in NGU as the global coordinate of the agent location. The dashed line indicates coverage of a uniform random policy. Error bars indicate standard error of the mean, over 5 replicas. See Appendix Table S4 for absolute coverage numbers.

geNet embeddings hurt exploration in the `find` task, where agents encounters more complex scenes (Figure 7b). By contrast, the language-shaped representations are well-suited for not only describing simple objects, but also have capacity for multi-object, complex scenes. Of course, current CLIP-style models can be further improved in their ability to understand multi-object scenes, which may explain why the benefits are less pronounced for the `find` task. However, as the performance of pretrained vision-language models improve, we expect to see those benefits transfer to this method and drive even better exploration.

## 6 Discussion

We have shown that language abstractions and pretrained vision-language representations improve the sample efficiency of existing exploration methods. This benefit is seen across on-policy and off-policy algorithms (Impala and R2D2), different exploration methods (RND and NGU), different 3D domains (Playroom and City), and various task specifications (lifting/putting, searching, and intrinsically motivated navigation). Furthermore, we carefully designed control experiments to understand how language contributes to better exploration. Our results are consistent with cognitive perspectives on human language—language is powerful because it groups together situations according to semantic similarity. In terms of the desiderata that Burda et al. [6] present, language is both compact and sufficient. Finally, we note that using pretrained vision-language representations to embed image observations enables more effective exploration even if language is not available during agent training. This is vital for scaling to environments that do not have a language oracle or annotations.

**Limitations and future directions** We highlight several avenues for extending our work. First, additional research could provide a more comprehensive understanding of how language abstractions affect representations. This could include comparing different types of captions offering varying levels of detail, or task-dependent descriptions. These captions could be dynamically generated at scale by prompting a large multimodal model [2]. Second, it would be useful to investigate how to improve pretrained vision-language representations for exploration by finetuning on relevant datasets. The semantics of a dataset could even be tailored to task-specific abstractions to increase the quality of the learnt representations. Such approaches would potentially allow applying our method to virtual

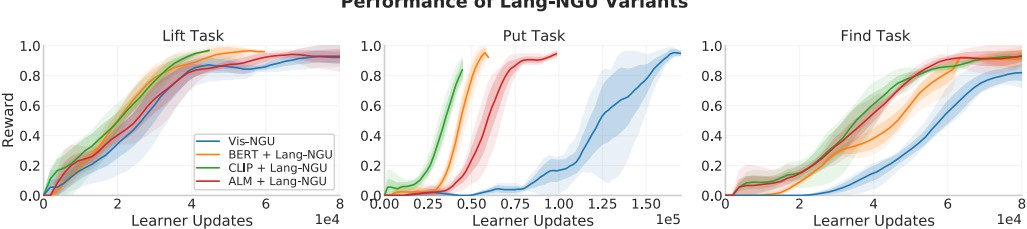

(a) Lang-NGU rewards novel pretrained text embeddings of $O_L$.

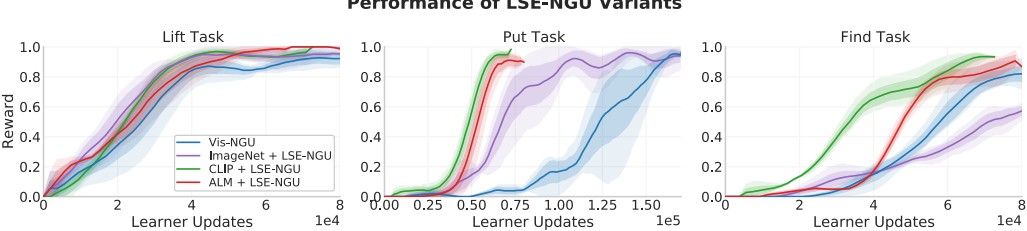

(b) LSE-NGU rewards novel pretrained image embeddings of $O_V$ and does not use oracle language.

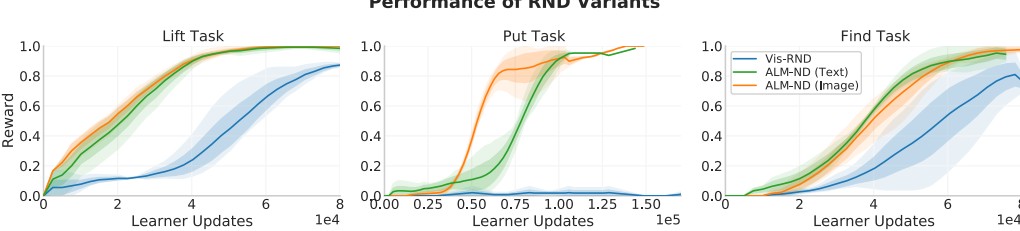

(c) ND intrinsic rewards derive from the prediction error of the representations from a pretrained ALM network.

Figure 7: Agents that use pretrained language-shaped representations to explore (ALM-ND, Lang-NGU, LSE-NGU) learn faster than baseline agents. ALM-ND (Text/Image) refer to the ND variants in Table 2. Results shown with a 95% confidence interval.

environments that are farther from the pretraining distribution, such as Atari. In contrast, compared to our experiments, we believe that the current pretrained representations would deliver even more benefit for entirely photorealistic, visually rich environments, such as Matterport3D [9]. Finally, we note that a limitation of this approach is that current pretrained vision-language models may be less effective on multi-object scenes. Future pretraining innovations or larger models would presumably produce more robust representations and thus lead to even more effective exploration.

## Acknowledgments and Disclosure of Funding

We would like to thank Iain Barr for ALM models and Nathaniel Wong and Arthur Brussee for the Playroom environment. For the City environment, we would like to thank Nick Young, Tom Hudson, Alex Platonov, Bethanie Brownfield, Sarah Chakera, Dario de Cesare, Marjorie Limont, Benigno Uria, Borja Ibarz and Charles Blundell. Moreover, for the City, we would like to extend our special thanks to Jayd Matthias, Jason Sanmiya, Marcus Wainwright, Max Cant and the rest of the Worlds Team. Finally, we thank Hamza Merzic, Andre Saraiva, and Tim Scholtes for their helpful support and advice.

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
