# Appendix

## A  Training Details

We use a distributed RL training setup with 256 parallel actors. For Impala agents, the learner samples from a replay buffer that acts like a queue. For R2D2 agents, the learner samples from a replay buffer using prioritized replay. Training took 8-36 hours per experiment on a $2 \times 2$ TPUv2.

All agents share the same policy network architecture and hyperparameters (Table S1). We use an Adam optimizer for all experiments. The hyperparameters used to train Impala [17] and R2D2 [25] are mostly taken from the original implementations. For Impala, we set the unroll length to 128, the policy network cost to 0.85, and state-value function cost to 1.0. We also use two heads to estimate the state-value functions for extrinsic and intrinsic reward separately. For R2D2, we set the unroll length to 100, burn-in period to 20, priority exponent to 0.9, Q-network target update period to 400, and replay buffer size to 10,000.

Table S1: Common architecture and hyperparameters for all agents.

| Setting | Value |
|---|---|
| Image resolution: | 96x72x3 |
| Number of action repeats: | 4 |
| Batch size: | 32 |
| Agent discount $\gamma$: | 0.99 |
| Learning rate: | 0.0003 |
| ResNet num channels (policy): | 16, 32, 32 |
| LSTM hidden units (memory): | 256 |

### A.1  Training Details for NGU Variants

We use a simplified version of the full NGU agent as to focus on the episodic novelty component. One major difference is that we only learn one value function, associated with a single intrinsic reward scale $\beta$ and discount factor $\gamma$. The discount factor $\gamma$ is 0.99 and we sweep for $\beta$ (Table S2). Another major difference is that our lifetime novelty factor $\alpha$ is always set to 1.

The NGU memory buffer is set to 12,000, so that it can always has the capacity to store the the entire episode. The buffer is reset at the start of the episode. The intrinsic reward is calculated from a kernel operation over state representations stored in the memory buffer. We use the same kernel function and associated hyperparameters (e.g. number of neighbors, cluster distance, maximum similarity) found in Badia et al. [3].

For Vis-NGU, the 32-dimension controllable states come from a learned inverse dynamics model. The inverse dynamics model is trained with an Adam optimizer (learning rate 5e-4, $\beta_1 = 0.0$, $\beta_2 = 0.95$, $\epsilon$ is 6e-6). For the variants, we use frozen pretrained representations from BERT, ALM, or CLIP. The ALM pretrained embeddings are size 768 and the CLIP embeddings are 512. Med-ALM comprises a 71M parameter NFNet image encoder and 77M BERT text encoder. Small-ALM comprises a 25M Resnet image encoder and 44M BERT text encoder. Using Small-ALM can help mitigate the increased inference time. To manage training time, we use Small-ALM in the City environment, where the episode is magnitudes longer than Playroom. For the LSE-NGU ImageNet control, we use the representations from a frozen 71M parameter NFNet pretrained on ImageNet (F0 from Brock et al. [5]). This roughly matches the size of the CLIP and Med-ALM image encoders.

We notice that it is crucial for the pretrained representations to only be added to the buffer every 8 timesteps. Meanwhile, Vis-NGU adds controllable states every timestep, which is as or more effective than every 8. This may be due to some interactions between the kernel function and the smoothness of the learned controllable states. We also find that normalizing the intrinsic reward, like in RND, is helpful in some settings. We use normalization for Lang-NGU and LSE-NGU on the Playroom tasks.

Table S2: Hyperparameters for the family of NGU agents on the Playroom tasks. All agents except Lang-NGU use a scaling factor of 0.01 in City. Lang-NGU uses 0.1.

| Environment | Method | Embedding Type | Intrinsic Reward Scale $\beta$ | Entropy cost |
|---|---|---|---|---|
| Playroom `lift`, `put` | Vis-NGU | Controllable State | 3.1e-7 | 6.2e-5 |
| | Lang-NGU | BERT | 0.029 | 2.4e-4 |
| | | $CLIP_{text}$ | 0.02 | 2.3e-4 |
| | | $ALM_{text}$ | 0.0035 | 9.1e-5 |
| | LSE-NGU | $CLIP_{image}$ | 0.029 | 2.6e-4 |
| | | $ALM_{image}$ | 0.012 | 1.6e-4 |
| | | ImageNet | 0.0072 | 6.4e-5 |
| Playroom `find` | Vis-NGU | Controllable State | 3.1e-6 | 2.6e-5 |
| | Lang-NGU | BERT | 0.029 | 2.4e-4 |
| | | $CLIP_{text}$ | 0.0047 | 4.1e-5 |
| | | $ALM_{text}$ | 0.013 | 1.9e-4 |
| | LSE-NGU | $CLIP_{image}$ | 0.0083 | 6.7e-5 |
| | | $ALM_{image}$ | 0.0051 | 1.2e-4 |
| | | ImageNet | 0.013 | 1.0e-4 |

## A.2 Training Details for RND-Inspired Agents

For the RND-inspired exploration agents, the hyperparameters for training the trainable network are taken from the original implementation [7]. We perform a random hyperparameter search over a range of values for the intrinsic reward scale, V-Trace entropy cost, and the learning rate for the trainable network. The values can be found in Table S3. We also normalize all intrinsic reward with the rolling mean and standard deviation.

Table S3: Hyperparameters for the family of RND-inspired agents. Learning rate is used for the trainable network.

| Environment | Setting | Intrinsic reward scale $\beta$ | Entropy cost | Learning rate |
|---|---|---|---|---|
| Playroom `lift`, `put` | Vis-RND | 1.4e-4 | 1.2e-4 | 1.2e-3 |
| | Lang-RND | 3.2e-6 | 8.1e-5 | 5.3e-4 |
| | ALM-ND (Text) | 8.4e-6 | 2.5e-5 | 3.1e-3 |
| | ALM-ND (Image) | 1.1e-5 | 2.2e-5 | 2.1e-3 |
| | LD | 1.1e-5 | 2.7e-5 | 1.7e-3 |
| | S-LD | 1.4e-6 | 7.6e-5 | 1.8e-4 |
| Playroom `find` | Vis-RND | 9.9e-5 | 4.4e-5 | 5.4e-4 |
| | Lang-RND | 7.2e-4 | 8e-5 | 1e-3 |
| | ALM-ND (Text) | 2.3e-4 | 3.9e-5 | 9.6e-4 |
| | ALM-ND (Image) | 3e.0-3 | 9.5e-5 | 2.1e-4 |
| | LD | 4.1e-5 | 4.3e-5 | 2e-3 |
| | S-LD | 4.1e-5 | 4.3 e-5 | 2.e-3 |

The various RND-inspired agents differ in the input and output space of the trainable network and target functions (Figure S1). Some trainable networks and target networks involve a convolutional network, which consists of (64, 128, 128, 128) channels with (7, 5, 3, 3) kernel and (4, 2, 1, 1) stride. For ND, we balance the model capacity of the trainable network with the memory required for learning larger networks. We notice that using larger networks as the target function can make distillation harder and therefore, requires more careful parameter tuning. Our experiments use the 26M parameter Small-ALM vision encoder for generating target outputs, which minimizes the discrepancy in complexity between the trainable network and target function. This also managed the memory requirements during training.

## A.3 Engineering Considerations

Integrating large pretrained models into RL frameworks is nontrivial. High quality pretrained models are orders of magnitudes larger than policy networks, introducing challenges with inference speed.

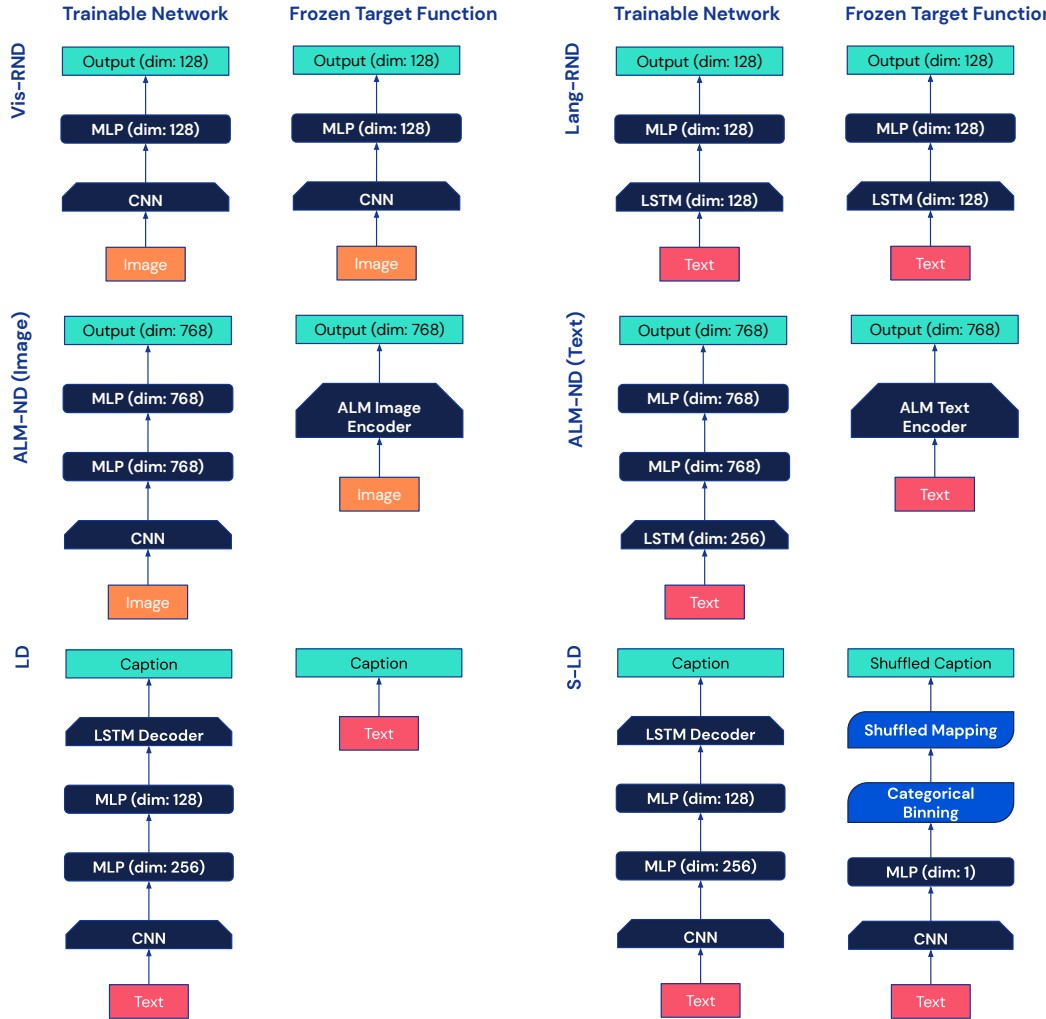

Figure S1: Architecture diagrams of the trainable network and frozen target functions for the RND-inspired family of methods. The teal square boxes are paired outputs, such that the trainable function is trained to match the frozen target function. None of the parameters here are shared with the policy or value network.

Slow inference not only increases iteration and training times but also may push learning further off-policy in distributed RL setups [e.g. 17, 25]. We mitigate this issue in Lang-NGU and LSE-NGU by only performing inference computations when it is necessary. We compute the pretrained representations only when they are required for adding to the NGU buffer (i.e. every 8 timesteps).

## A.4 S-LD Construction

As explained in Section 5.1, we compare the efficacy of exploration induced by the language distillation (LD) method with that induced by a shuffled variant, S-LD. LD employs an exploration bonus based on the prediction error of a captioning network $f_C : O_V \to O_L$, where the target value is the oracle caption. Our goal with S-LD is to determine whether the efficacy of the exploration is due to the specific abstractions induced by $f_C$, or whether it is due to some low level statistical property (e.g. the discrete nature of $O_L$, or the particular marginal output distribution).

We sample a fixed, random mapping $\hat{f}_S : O_V \to O_L$ while trying to match the low-level statistics of $f_C$. To do this, we construct a (fixed, random) smooth partitioning of $O_V$ into discrete regions, which in turn are each assigned to a unique caption from the output space of $f_C$. The regions are sized to maintain the same marginal distribution of captions, $P_{\pi_{LD}}(L) \approx P_{\pi_{S-LD}}(L)$.

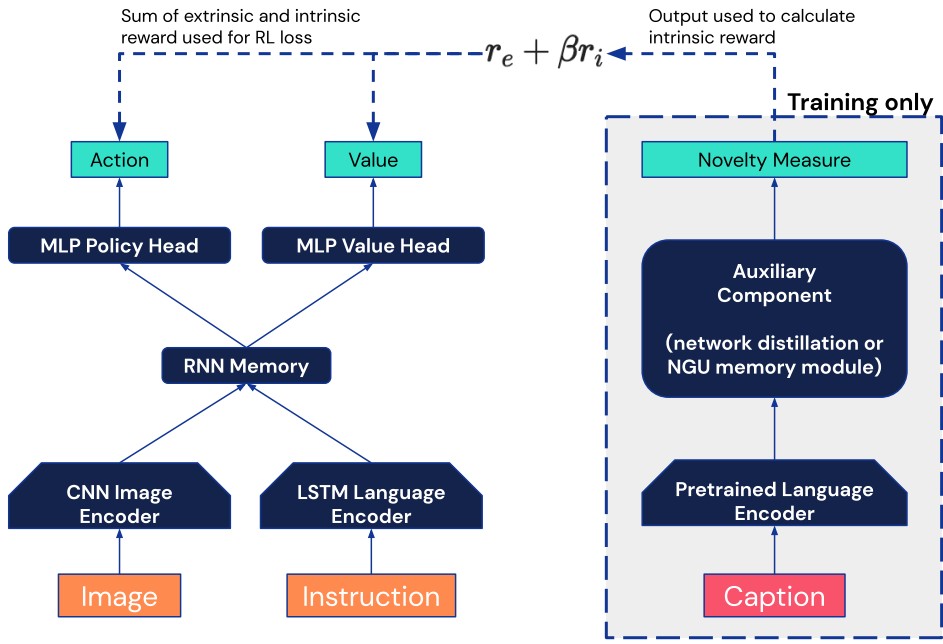

Figure S2: Architecture diagram for a generic Impala agent used in our Playroom experiments. We feed the image observation and language instruction to the policy and value networks. During training, we also use the scene caption to calculate the intrinsic reward, which corresponds to the gray shaded box. There is no parameter sharing between the networks inside and outside of the gray box.

More precisely, our procedure for constructing $\hat{f}_S$ is as follows. We build $\hat{f}_S$ as the composition of three functions, $\hat{f}_S = g_3 \circ g_2 \circ g_1$:

- $g_1 : O_V \to \mathbb{R}$ is a fixed, random function, implemented using a neural network as in Vis-RND.

- $g_2 : \mathbb{R} \to Cat(K)$ is a one-hot operation, segmenting $g_1(O_V)$ into $K$ non-overlapping, contiguous intervals.

- $g_3 : Cat(K) \to O_L$ is a fixed mapping from the discrete categories to set of captions produced by the language oracle, $f_C$.

To ensure that the marginal distribution of captions seen in $f_C$ was preserved in $f_S$, we first measure the empirical distribution of oracle captions encountered by $\pi_{LD}$, a policy trained to maximize the LD intrinsic reward. We denote the observed marginal probability of observing caption $l_i$ as $P_{\pi_{LD}}(O_L = l_i) = q_i$ (where the ordering of captions $l_i \in O_L$ is fixed and random). We then measure the empirical distribution of $g_1$ under the same action of the same policy, $P_{\pi_{LD}}(g_1(O_V))$, and define the boundaries of the intervals of $g_2$ by the quantiles of this empirical distribution so as to match the CDF of $\mathbf{q}$, i.e. so that $P_{\pi_{LD}}(g_2 \circ g_1(O_V) = i) = q_i$. Finally, we define $g_3$ to map from the $i^{\text{th}}$ category to the corresponding caption $l_i$, so that $P_{\pi_{LD}}(g_3 \circ g_2 \circ g_1(O_V) = l_i) = q_i$.

## B  Additional ablation: Pretrained controllable states

The state representations used by Vis-NGU are trained jointly with the policy, whereas the pretrained representations used by Lang-NGU and LSE-NGU are frozen during training. To isolate the effect of the knowledge encoded by the representations, we perform an additional experiment where we pretrain the controllable states and freeze them during training. We use the weights of the inverse dynamics model from a previously trained Vis-NGU agent. Figure S3 shows that pretrained Vis-NGU learns at the same rate as Vis-NGU (if not slower). Thus, the increased performance in Lang-NGU and LSE-NGU agents is due to the way the vision-language embeddings are pretrained on captioning datasets. This furthermore suggests that the converged controllable states do not fully

capture the knowledge needed for efficient exploration and in fact may even hurt exploration at the start of training by focusing on the wrong parts of the visual observation.

Figure S3: Vis-NGU with a pretrained inverse dynamics model learns slower than the baseline Vis-NGU agent that uses online learned controllable states.

## C   Additional Figures

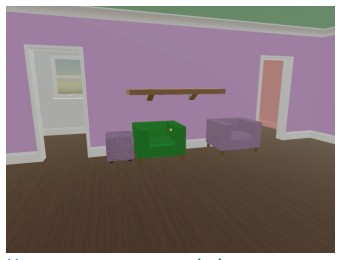

You can see two arm chairs, an ottoman, and a shelf.

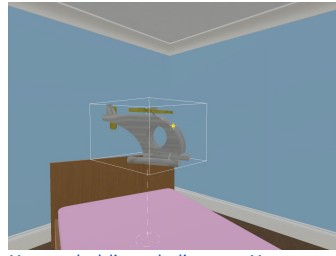

You are holding a helicopter. You can see a bed.

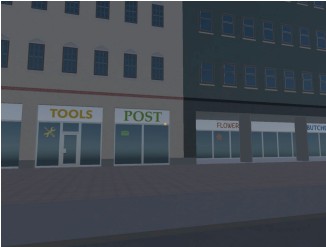

You are looking at a post office in a light grey stucco shops in a apartment building.

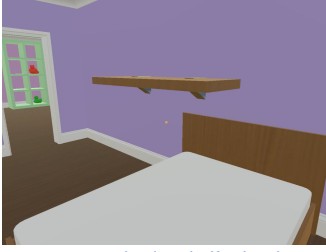

You can see a bed, a shelf, a bookcase, a teddy, and a rubber duck.

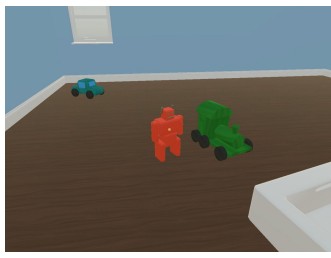

You can see a storage tray, a train, a robot, and a car.

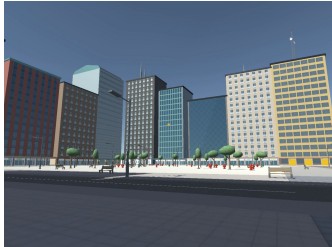

You are looking at a blue grey brick offices in a office building.

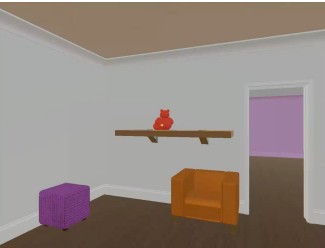

You can see an armchair, an ottoman, a shelf, and a teddy.

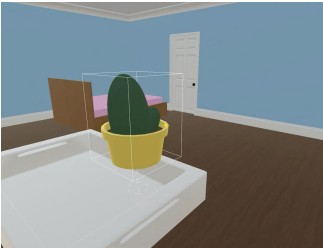

You are holding a potted plant. You can see a storage tray and a bed.

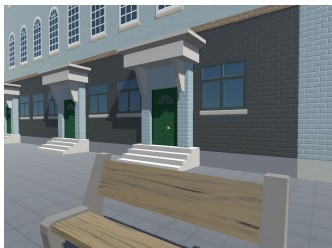

You are looking at a green door in a dark grey painted brick ground floor apartments in a apartment building.

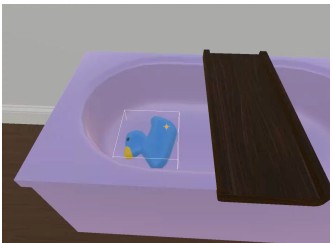

You can see a bathtub and a rubber duck.

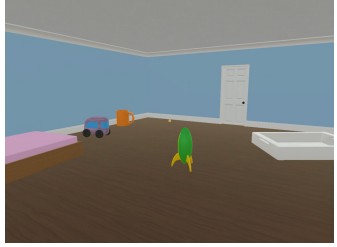

You can see a bed, a storage tray, a rocket, a bus, and a mug.

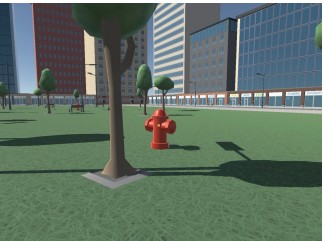

You are looking at a fire hydrant in a park.

Figure S4: Example scenes and associated captions from multi-room Playroom used for the `find` task (left column), single-room Playroom used for the `lift` and `put` tasks (middle column), and City (right column).

Table S4: Mean and standard error of coverage (number of bins reached on map) by variants of NGU agents using different state representations. The City consists of 1024 total bins, although not all are reachable. With the ground truth (continuous) embedding type, the NGU state representation is the global coordinate of the agent location. With the ground truth (discrete) embedding type, the representation is a one-hot encoding of the bins. A non-adaptive, uniform random policy is also included as baseline ('N/A - Random Actions').

| Embedding Type | Coverage (number of bins) |
|---|---|
| Ground Truth (continuous) | $346 \pm 3.2$ |
| Ground Truth (discrete) | $539 \pm 3.5$ |
| N/A – Random Actions | $60 \pm 0.53$ |
| NGU with Controllable State | $83 \pm 6.9$ |
| NGU with ImageNet | $111 \pm 10.6$ |
| Lang-NGU with CLIP | $225 \pm 8.9$ |
| Lang-NGU with Small-ALM | $241 \pm 9.0$ |
| LSE-NGU with CLIP | $153 \pm 7.1$ |
| LSE-NGU with Small-ALM | $162 \pm 5.6$ |

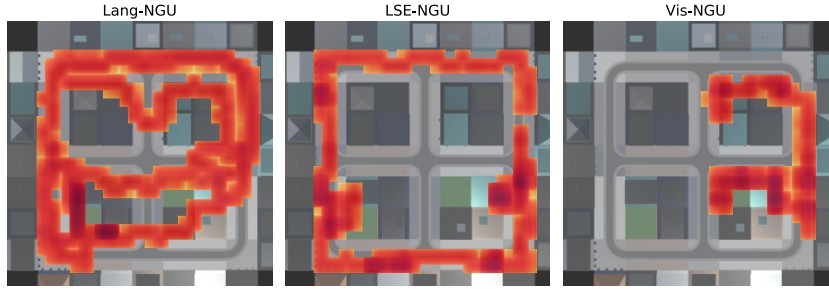

Figure S5: Heatmaps of agent coverage over the City environment.

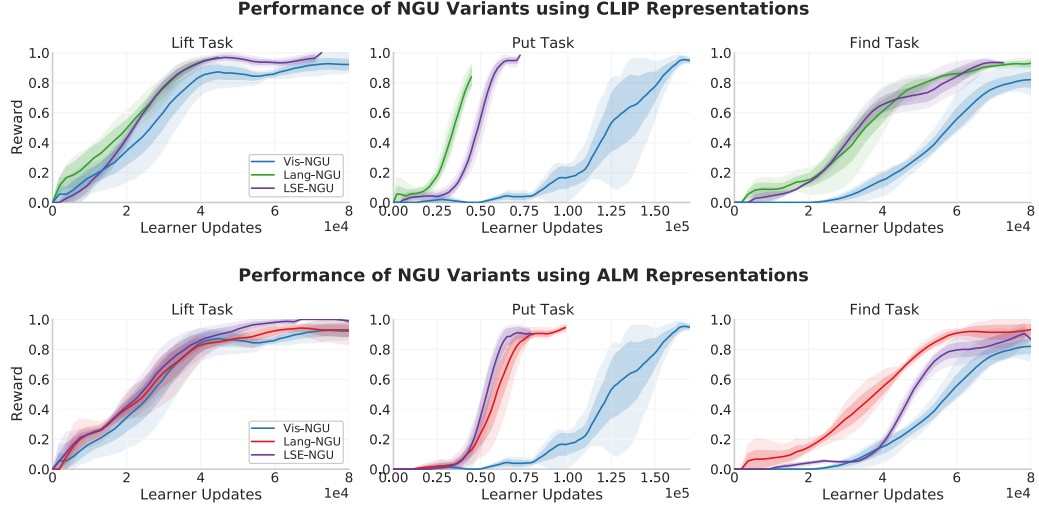

Figure S6: For both CLIP and ALM representations, LSE-NGU and Lang-NGU learn at comparable speeds, suggesting that the method can transfer well to environments without language annotations.

**Number of Objects Held by Lang-NGU Variants**

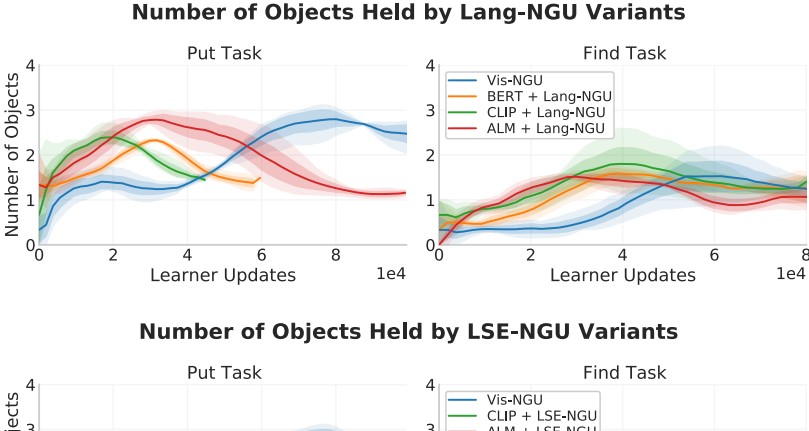

**Number of Objects Held by LSE-NGU Variants**

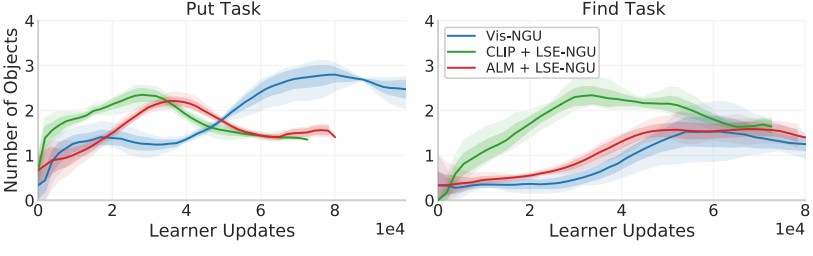

**Number of Objects Foveated by Lang-NGU Variants**

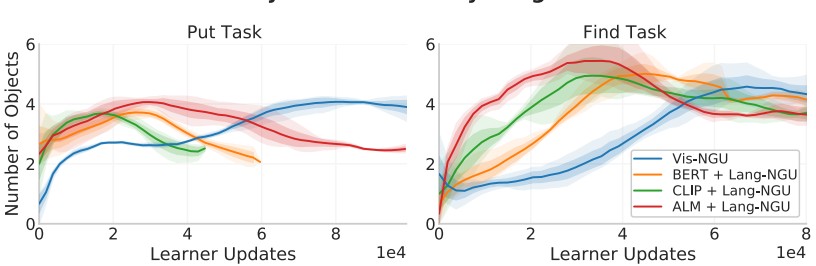

**Number of Objects Foveated by LSE-NGU Variants**

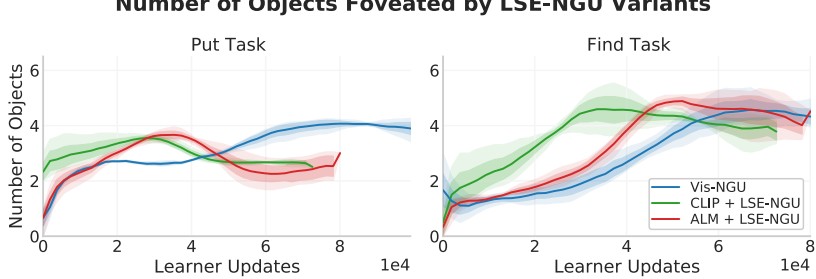

Figure S7: Lang-NGU and LSE-NGU agents learn to interact with objects (holding and foveating) earlier in training compared to the Vis-NGU agent. The benefit is larger for the `put` task, where the extrinsic reward also reinforces object interaction.