# OpenReview forum: "Semantic Exploration from Language Abstractions and Pretrained Representations"
_NeurIPS.cc/2022/Conference — NeurIPS 2022 Accept_

### Official Review · Reviewer_jixR · 2022-06-19

**Rating:** 5
**Confidence:** 3
**Soundness:** 3 good
**Presentation:** 3 good
**Contribution:** 2 fair

**Summary:**

To guide RL algorithms to explore novel states, the paper proposes to leverage language based representations. From Burda, language is both compact and sufficient, thus giving strong motivations for such approaches. Authors first successfully show that using scene description language as an oracle, allows the existing algorithms to achieve better performance for accomplishing tasks (Playroom) or perform better at exploration (City). Then the authors show that using caption pretrained networks can also achieve better performance than relying on visual features alone in both environments. Technically, the authors incorporate language (or language motivated representations) into NGU and RND. In both cases, the representation is used to calculate the intrinsic rewards for the corresponding RL algorithm.

**Questions:**

Did the authors further explore ND type of training ? For example, we can think of experiments such as generating captioning by a pretrained model and conditioned from the language? Such experiments might shed light on how useful language really is as a representation? Or maybe the likelihood loss is generally inappropriate when we apply end to end training? Can authors share some thoughts on this aspect please?

How effective can the language be in an end-to-end training? For example, if I give a language oracle for just several scenes, what should be the design to leverage these representations to make the model better when language is no more present at test time? I agree thatt this is slightly off-topic for the paper’s main point.

I have two minor comments for the paper that I put in this Question section:
- I feel that mentioning Impala and R2D2 explicitly is not necessary in the abstract; after all, the paper proposes a general method that is expected to work across environments and algorithms.
- The acronyms RND and NGU introduced in the introduction looks confusing, especially they don’t correspond to acronyms that can be deduced from the text.

**Limitations:**

The limitation section in this paper lists quite some important directions and items. In addition, as I listed in the strength/weakness section, to seek what this paper brings in particular compared to other approaches to integrate text inspired representations; for example, what is the advantage to integrate it in intrinsic reward instead of solely changing the representations, etc.

**Strengths And Weaknesses:**

Strength

Although Khandelwa et al. 2021 have shown that CLIP pretrained representations can greatly help goal oriented RL tasks, this paper goes one step further showing that NL representation is a very effective representation and might explain from another angle the effectiveness of methods such as CLIP. The investigation is thorough by first showing the NL representation upper bound; the proposed method has been tested using two datasets with different frameworks.

Weakness

For me, the paper hasn’t shown comparisons with other pretrained model representations, thus it is very difficult to measure the paper’s contribution compared to the SOTA method. For example, line 103, the paper indicates that their method is complementary to Khandelwa et al. 2021 and thus can combine with to achieve better performance. However, given both methods capture some semantic representation about the environment, it is not clear why a better performance can even be expected? Maybe the proposed method captures this information better than Khandelwa et al. 2021? Maybe it is the inverse? Without experiments, all the above questions remain unanswered, making it difficult to situate the paper in its area.

---

> ### Author Response · Authors · 2022-08-02
> **Rebuttal for Reviewer jixR**
>
> > For me, the paper hasn’t shown comparisons with other pretrained model representations, thus it is very difficult to measure the paper’s contribution compared to the SOTA method. For example, line 103, the paper indicates that their method is complementary to Khandelwa et al. 2021 and thus can combine with to achieve better performance. However, given both methods capture some semantic representation about the environment, it is not clear why a better performance can even be expected? Maybe the proposed method captures this information better than Khandelwa et al. 2021? Maybe it is the inverse?
>
> Khandelwa et al. propose a method to improve perception in an embodied agent by using CLIP embeddings. Their work is not a method for better intrinsic motivation or exploration in agents - so it is not directly relevant to this research. We mentioned the work because it is an example of using CLIP representations for embodied AI - but it is not directly relevant to the research discussed in our paper.
>
> To be more specific, in Khandelwa et al.'s agent, CLIP model weights are used in the visual perception of an agent. In contrast, we use CLIP to derive an intrinsic reward signal to stimulate exploration. Unlike Khandelwa et al, our agent is not conditioned on CLIP representations (i.e. they are not an input to our agent), and (also unlike Khandelwa et al) we do not use CLIP at all when evaluating our agent (only for training). This confers our method with the ability to use larger pretrained models, as the policy network remains small and inference time is minimal.
>
> Of course, it would be possible to apply our method with a policy that includes CLIP representations for visual perception, but doing so would be quite orthogonal to the issue at hand - namely determining good representations to drive intrinsic motivation. This was the point that we were trying to make in the original discussion, but we agree it was misleading. We will make all of this extra clear in the camera-ready version, taking advantage of the extra page. We also clarify and better situate how our work complements prior methods in the related works section.
>
> > Did the authors further explore ND type of training ? For example, we can think of experiments such as generating captioning by a pretrained model and conditioned from the language? Such experiments might shed light on how useful language really is as a representation? Or maybe the likelihood loss is generally inappropriate when we apply end to end training? Can authors share some thoughts on this aspect please?
>
> We experimented with the ND setting as a way to understand the upper limit of how language can be helpful, in this case if we have access to oracle captions. We did not (as you suggest) try replacing this oracle with a pretrained caption-generation system, because the oracle gave us a more controlled understanding of this upper bound. Instead, we focused on using CLIP's (text-aligned) visual embedding, which is much more computationally efficient (we do not need to sample from the model during training) and resulted in good performance.
>
> We note that CLIP's image embeddings are implicitly shaped by language during the pretraining process. We compare those representations to ImageNet and find that it performed significantly better, which supports our view that language is playing an important role in strong representations for exploration (Figure 6B).
>
> > How effective can the language be in an end-to-end training? For example, if I give a language oracle for just several scenes, what should be the design to leverage these representations to make the model better when language is no more present at test time? I agree thatt this is slightly off-topic for the paper’s main point.
>
> We want to clarify that language captions and representations thereof are only used to guide exploration via intrinsic reward during training. They are not used as input to the policy. Captions are not required at test time, and network distillation methods do not involve any end-to-end training between policy and auxiliary distillation networks. We’ve now added additional clarification and visualizations (Figure S2) as we feel like this point isn’t being emphasized enough in the main text, as evidenced by confusion expressed by multiple reviewers.
>
> > I feel that mentioning Impala and R2D2 explicitly is not necessary in the abstract; after all, the paper proposes a general method that is expected to work across environments and algorithms.
>
> This is a good point, we’ve now modified the abstract to highlight that the particular specifics of these agents are not important for this method to work, thanks for the suggestion!
>
> > The acronyms RND and NGU introduced in the introduction looks confusing, especially they don’t correspond to acronyms that can be deduced from the text.
>
> We apologize for not defining these acronyms and have now updated our paper to do this upon the first use.

---

### Official Review · Reviewer_kb5q · 2022-07-08

**Rating:** 8
**Confidence:** 4
**Soundness:** 4 excellent
**Presentation:** 4 excellent
**Contribution:** 4 excellent

**Summary:**

In this paper, the authors propose to use language to meaningfully abstract the observable state of an environment. The authors argue this language abstraction can be used to improve exploration when training reinforcement learning agents. Specifically, the authors propose to use the text description/caption of a state to provide an intrinsic reward to the agent for exploring new states that are semantically different from each other. To test language-based exploration, the authors use two popular exploration methods, Never Give Up (NGU) and Random Network Distillation (RND), and augment them with language. Experiments were conducted on both on-policy (Impala) and off-policy (R2D2) algorithms on several goal-oriented tasks within two environments Playroom and City. A first set of experiments was performed with a language oracle captioner as a proof-of-concept, then a second set that uses pretrained vision-language models to provide captions instead of the oracle. Empirically, it was shown that language-based exploration yields faster convergence, and greater state-space coverage across the different exploration methods and RL algorithms in the tested environments.


**Questions:**

- While I agree with the following claim "The human-generated captions structure the visual embedding space to reflect features most pertinent to humans and human language.", do the authors have a reference for it?
- Is the code to reproduce the results will be made available?

**Limitations:**

The authors discussed the limitations of the proposed approach, notably that vision-language models are susceptible to domain shift and less effective on multi-object scenes. Both limitations are active areas of research in the computer vision community which would benefit indirectly the proposed approach. The authors do not mention the social impacts of their research. I believe the proposed technique shares similar social impacts as any other goal-oriented RL agents, e.g. unsafe behaviors during exploration.


**Strengths And Weaknesses:**

**What I like about this paper**
- Using language as a useful abstraction that coarsens the state space while preserving the semantics of the environment.
- Using a language oracle as a proof-of-concept, then showing promising results when access to such oracle is not possible and have to rely on vision-language models as captioners.
- The proposed technique is modular and can be applied to different RL algorithms (both on-policy and off-policy) and exploration techniques.
- Language models are being used out-of-the-box without any finetuning on the tested environments. I like the reusability aspect of it.

**Potential weaknesses**
- Language abstraction of a state should take into account the current goal. For instance, some details about an object might be relevant in one task but not in others. That said, the authors are aware of this and allude to it in the future directions section.
- Regarding the claim about random TV noise. While I agree that language could avoid getting stuck in front of a TV with random static noise, it is still susceptible to random text/images appearing on the TV (provided the language oracle describes it). While it might be an easier issue to deal with, I would still mention it has a limitation: how do you determine the 'correct' level of abstraction?
- It is not clear how stable the vision-language models (or even the language oracle) are when generating text for two adjacent frames (i.e., a slight modification of the image). For instance, does the order in which the objects in a scene are described change?

**Originality, quality, clarity, and significance**

To the best of my knowledge, the work in this paper is novel. In particular how the authors proposed to augment NGU and RND exploration methods with language. The authors do mention concurrent work and how the proposed approach differs from theirs. The paper is clearly written, well-organized, and technically sound to me. All claims are empirically backed up by experiments. With the recent advances in language modeling and computer vision, I believe the proposed technique is of interest to the research community. The modular aspect of language-based exploration should help drive wider adoption.

Overall, I recommend this paper for acceptance.

---

> ### Author Response · Authors · 2022-08-02
> **Rebuttal for Reviewer kb5q**
>
> We’re gratified at the reviewer’s positive response and for their highly useful comments.
>
> > Language abstraction of a state should take into account the current goal. For instance, some details about an object might be relevant in one task but not in others. That said, the authors are aware of this and allude to it in the future directions section.
>
> This is a good point and we agree. An example of how this could be implemented is to use a pretrained multi-modal model, such as Flamingo. Instead of using the representations from a single modality, we could take the image embedding produced by the pretrained model while prompting it with a language description of the goal. We will further explore these ideas in the discussion of the camera-ready version.
>
> > Regarding the claim about random TV noise. While I agree that language could avoid getting stuck in front of a TV with random static noise, it is still susceptible to random text/images appearing on the TV (provided the language oracle describes it). While it might be an easier issue to deal with, I would still mention it has a limitation: how do you determine the 'correct' level of abstraction?
>
> Great point! One of the main motivations of using natural language as the basis of the novelty signal is that we believe it provides just the “right” level of abstraction, since language is typically generated within a specific context for a specific purpose. One can imagine that a pixel-by-pixel language description of an image would not be useful at all. Our experiments with the language oracle used a generic level of abstraction that describes the objects in the scene, which worked well with the types of tasks we were interested in (Figure S4 for example captions). The pretrained vision-language models similarly contain a useful level of abstraction in that the pretraining dataset comprises captions describing common, 3D scenes containing everyday objects. A further point of enquiry is to investigate how the dataset on which the pretrained models were trained affects downstream performance, as the dataset can be curated for the preferred level of detail.
>
> We will add these points to the camera-ready version, taking advantage of the extra page.
>
> > It is not clear how stable the vision-language models (or even the language oracle) are when generating text for two adjacent frames (i.e., a slight modification of the image). For instance, does the order in which the objects in a scene are described change?
>
> Our language oracle tries to reflect a perfect state abstraction, and thus is stable and does not change the order in which objects in a scene are described. With the vision-language models, the text embeddings of similar captions “You can see a dog and cat” versus “You can see a cat and dog” are not identical but are relatively close in the embedding space. However, both captions would be far from “You can see a boat.” This provides some amount of robustness to any imperfectly generated captions. Likewise, the pretrained vision embeddings will not be perfectly identical for adjacent frames but provide a robust approximation of semantically similar state abstractions.
>
> > While I agree with the following claim "The human-generated captions structure the visual embedding space to reflect features most pertinent to humans and human language.", do the authors have a reference for it?
>
> We have now included a reference in the updated paper.
>
> > Is the code to reproduce the results will be made available?
>
> We will open source the code upon acceptance.

---

> > ### Comment · Reviewer_kb5q · 2022-08-05
> > **Response to authors' rebuttal**
> >
> > Thank you for the rebuttal and for providing your insights on the questions I raised.
> >
> > To me, this paper showcases empirical evidence of successfully leveraging state abstraction (here using language) to provide intrinsic exploration reward compared to using random and entropy exploration techniques (which are still the go-to techniques for the majority of RL works). I'm hopeful the research community will build on it to push SOTA on different embodied environments and work on theoretical guarantees for such an approach.

---

### Official Review · Reviewer_mRW6 · 2022-07-10

**Rating:** 4
**Confidence:** 3
**Soundness:** 2 fair
**Presentation:** 2 fair
**Contribution:** 2 fair

**Summary:**

This paper proposes to use semantic representation of the visual environment to guide the agent to explore novel scenes in 3D tasks.
 Either oracle description of the environment or vision-and-text representation from a large language model can be used to calculate reward to guide a Reinforcement Learning based agent. They validate the idea via implementing Impala on Playroom and R2D2 on City. The results are promising. However, many details about the models and the experiments are missing. The authors mention appendix many times but I did not find it.

**Questions:**

N/A

**Strengths And Weaknesses:**

Strengths

1. The idea of using language as a guidance to encourage the agent to explore more is interesting. As is discussed in the paper, language is abstract and compact, making it great cue for the environment information.

2. The language oracle can be replaced by a vision-and-language representation.

2. Successfully incorporating it into the RL system is also important. They demonstrated how to do it in two agents.

3. Good experimental results in two different environments.

Weakness:
1. The authors are using Unity, meanwhile more 3D navigation tasks are using photo-realistic environment such as Matterport-3D.

2. Lack of theory support. For example, how the language actually influences the exploration process.

3. Missing a lot of details. I guess some details might be in the missing appendix file.

---

> ### Author Response · Authors · 2022-08-02
> **Rebuttal for Reviewer mRW6**
>
> > However, many details about the models and the experiments are missing. The authors mention appendix many times but I did not find it.
> > Missing a lot of details. I guess some details might be in the missing appendix file.
>
> We are incredibly sorry to have omitted the appendix - this was entirely unintentional. We hope that having access to the appendix clears up a lot of the confusion regarding our methods and experiments.
>
> > The authors are using Unity, meanwhile more 3D navigation tasks are using photo-realistic environment such as Matterport-3D.
>
> The reason that we use Unity in this work is that we wish to study 'semantic' tasks that require the recognition and manipulation of 3D objects (much as in the real world). While Matterport-3D is photorealistic, it is impossible to interact with objects in Matterport environments. Moreover, exploration challenges are simpler in Matterport-3D, which are based on discrete state spaces. In contrast, all positions, actions and other physical quantities like momentum vary continuously in our environments, making exploration substantially more challenging.
>
> We also want to underline that our environments are visually rich, if not photorealistic. The City environment even simulates shadows and realistic lighting changes, resulting from sunrise and sunset events that occur in the episode. We refer the reviewer to Supplementary Figure S3 for example screenshots of the environment.
>
> Nonetheless, we expect that our method would perform even better in entirely photorealistic environments like Matterport3D, for example because the scenery would align better with the types of images used to pretrain the vision-language models (realistic photographs scraped from the internet). Furthermore, we expect that such visually rich and detailed environments would benefit even more from the kind of semantic abstraction that comes from language, leading to even larger discrepancies between pixel- and language-based exploration strategies. We will add these points to our discussion section for the camera-ready, taking advantage of the extra page.
>
> > Lack of theory support. For example, how the language actually influences the exploration process.
>
> As the reviewer notes, our hypothesis is that language is semantically abstract and compact, and thus can move the exploration process towards more meaningfully different states. The theoretical underpinnings, like those of the majority of deep-learning methods in ML, are an open question. Deriving an analytic theory for language-driven exploration is simply beyond the scope of this work.
>
> We instead strive to empirically demonstrate the benefits of language with thorough, controlled empirical experiments, such as the S-LD comparisons (see Figures 3-4 and Appendix Section A.4). We compare ND using aligned language captions versus ND using shuffled/misaligned language captions. Although the misaligned captions preserve the coarseness of the state space, agents explore far less effectively. This experiment was the result of substantial thought and multiple iterations of design and redesign. The results support our hypothesis that language improves novelty-based exploration by grouping semantically similar, yet visually different states. Our other ablation experiments with ImageNet further elucidate the specific role that language plays in encouraging novelty-based exploration (eg Figure 6B).

---

> > ### Author Response · Authors · 2022-08-09
> > **Any Remaining Questions from Reviewer mRW6?**
> >
> > Dear reviewer,
> >
> > Thank you for your time and questions. We hope that our response, edits, and appendix highlight the work's contribution and clarify experimental details. Please let us know if you have additional questions. Alternatively, if you feel that your original concerns are addressed, we would appreciate updating your evaluation to reflect that.

---

### Official Review · Reviewer_p9aE · 2022-07-11

**Rating:** 6
**Confidence:** 3
**Soundness:** 3 good
**Presentation:** 2 fair
**Contribution:** 3 good

**Summary:**

The paper motivates the use of language abstractions instead of raw visual state representation for designing intrinsic exploration reward. As language can provide coarser and compact description of the environment, it can prevent unnecessary actions and meaningfully incentivize the agent to achieve the task. To this end, the paper proposes to use pre-trained models on image captioning datasets to enable better task-relevant exploration in 3D simulated environments. The experiments demonstrate that (1) random network distillation on oracle captions performs better than visual state representation and (2) semantics in the oracle caption matter, especially in longer horizon tasks. Furthermore, the use of ALM, CLIP or Imagenet embeddings based NGU or RND for exploration achieves faster task success as compared to using raw visual state representations.

**Questions:**

- The motivation why language enables wider exploration is unclear in Figure 1.
- Appendix is missing?
- In line 113, the policy receives … template-based “language” goal?
- What held-out goal settings, ie. in terms of unseen objects or environments considered in evaluation for different tasks?
- Can you clarify what ALM-ND (Text) and ALM-ND (Image) mean in Fig 6(c)? It seems that ALM-ND (X) is extracted from the pre-trained X encoder to use for network distillation and generate intrinsic reward.
- ‘Find’ task in Playroom seems very similar to the maximizing coverage in object navigation [1] Chaplot et al. How does the previous work compare as a baseline in terms of achieved coverage in held-out city environments?

**Ethics Review Area:**

["I don’t know"]

**Limitations:**

The authors provide detailed limitations and future scope, namely: (1) comparing different ways of captioning, (2) fine-tuning pre-trained VL models on relevant datasets for interaction tasks, and (3) having pretrained models on multi-object scenes to improve performance without oracle captions. Extending upon (1), the proposed approach seems to have an objective way of captioning any state that is relevant to solve the task effectively, whereas natural language captions are context-driven, underspecified in terms of all the objects in the seen and mostly describing main semantic takeaway. This assumption should be noted before using the proposed method to design intrinsic rewards for physical robots task.

**Strengths And Weaknesses:**

The paper is overall well-written and sound. The contribution in terms of experiments showing the value of using language based abstraction in modeling intrinsic reward in 3D scenes is novel and promising step towards exploration techniques for robot learning.
However, the success of using language based representations for exploration is not itself novel and somewhat depends on the kind of. captions and the task. This is reflected in Fig 6 and the authors also suggest experimenting with different captioning strategies as future research direction.

---

> ### Author Response · Authors · 2022-08-02
> **Rebuttal for Reviewer p9aE**
>
> We thank the reviewer for their detailed feedback, and are glad they found the contribution well-presented and the experiments relatively novel. We respond point-by-point to their comments below:
>
> > However, the success of using language based representations for exploration is not itself novel and somewhat depends on the kind of. captions and the task.
>
> We agree that the kind of representations that are optimal for exploration will always be task-dependent to some extent. Prior work with language and RL have built environment-specific parsers to achieve that type of representation. However, we think ours is the first study to show that pretrained vision-language models contain general enough representations that they are useful for novelty-based exploration in at least two very different, 3D first person tasks.
>
> > The motivation why language enables wider exploration is unclear in Figure 1.
>
> This figure gives intuition into how exploring based on language representations enables more extensive exploration than that based only on visual novelty. Semantically meaningful states are delineated by the navy dashed lines. If we use representations that reflect these boundaries (i.e. language), then an agent will more effectively explore the wider state space. If the representations do not reflect these boundaries and instead are amenable to visual noise (i.e. different colors, viewpoints, etc.), then an agent may only focus on a visually novel, yet novel subset of states (i.e. exploring only within a single localized patch). We thank the reviewer for raising this point. We have now clarified this caption in the paper.
>
> > In line 113, the policy receives … template-based “language” goal?
>
> Previous work uses hand-crafted environment parsers or human annotators to get captions that are specific to their environment/task. We’d like to highlight that our work is more general, because large-scale pretrained vision encoders are effective for 3D naturalistic environments. Furthermore, some prior methods also pass these templated captions to the policy to assist with training. Our method does not require captions or pretrained representations to be available for acting. We will clarify and highlight this difference in the camera ready.
>
> > What held-out goal settings, ie. in terms of unseen objects or environments considered in evaluation for different tasks?
>
> In both Playroom and the City, the environment procedurally generates a random configuration from an effectively infinite set of possibilities every episode. In Playroom, we randomly sample objects, object attributes, placements, room layouts, and language instructions. In the City experiments, we randomly sample a new map, with different buildings and building placements, and the agent must deal with changing lighting throughout the episode, as well as a random respawn point. This means that performance in unfamiliar scenarios is certainly reflected in our evaluation metrics for these two environments.
>
> Regarding unseen objects, the agent cannot handle new instructions that involve objects with unfamiliar names, because its encoder will not have been trained on those objects. The focus of this work is on how an agent can learn to explore better, rather than on that sort of visual-linguistic generalization.
>
> > Can you clarify what ALM-ND (Text) and ALM-ND (Image) mean in Fig 6(c)? It seems that ALM-ND (X) is extracted from the pre-trained X encoder to use for network distillation and generate intrinsic reward.
>
> This is correct. The difference between the two is the input to the pretrained encoder i.e. if we're distilling the text encoder or vision encoder. We have now clarified in the caption and added a reference to Table 2, in which this is defined. We also refer the reviewer to Table S1, which visually illustrates the differences between ALM-ND (Text/Image).
>
> > ‘Find’ task in Playroom seems very similar to the maximizing coverage in object navigation [1] Chaplot et al. How does the previous work compare as a baseline in terms of achieved coverage in held-out city environments?
>
> Compared to Chaplot, et al., our method is more general, because it does not rely on privileged state information (e.g. agent location). We consider novelty-based exploration that (as we show) is applicable to both navigation and object manipulation tasks. In contrast, Chaplot, et al. use SLAM-style models to construct an explicit map of the environment; it is not clear how their method would extend to object manipulation. When considering novelty-based algorithms, we focus on the specific question: “what is an enhanced representation for novelty?”. We have clarified the introduction section to make this much clearer in the revised version of the paper.

---

> > ### Author Response · Authors · 2022-08-09
> > **Any Remaining Questions from Reviewer p9aE?**
> >
> > Dear reviewer,
> >
> > Thank you for your time. We appreciate your questions and hope that our additional edits and figures improve the clarity of the work. Please let us know if you have additional questions. Alternatively, if you feel that your original concerns are addressed, we would appreciate updating your evaluation to reflect that.

---

### Author Response · Authors · 2022-07-27
**Appendix for the Paper**

We apologize for the error in omitting the appendix and any confusion this may have caused. It has been shared anonymously [here](https://drive.google.com/file/d/1Pq-6xKcskFT_S8XhuLkYpVaOSHO4jwLF/view?usp=sharing). We hope the reviewers find that it fills in a lot of the missing gaps. We plan to further address the reviewers’ comments but wanted to attach this as soon as possible in the rebuttal period.

---

### Author Response · Authors · 2022-08-02
**General Rebuttal**

We’d like to thank all reviewers for their thoughtful comments and feedback, and again apologize for the oversight in omitting the appendix. We have since uploaded it, and hope that the reviewers find that it greatly improves the clarity of our paper.

There were also several comments regarding lack of clarity in the text, and in response we have made substantial improvements to the revised version, and have committed to adding additional discussion points in the camera-ready version (detailed in reviewer-specific comments below).

Although the idea of language-based exploration has been proposed before, our contribution is a generally applicable way of implementing the idea in naturalistic first-person 3D environments using pretrained vision-language models.

Furthermore, we want to clarify a potential misconception: language captions are only used to guide exploration via intrinsic reward during training. Whereas prior methods may require annotations at test time, our work does not condition the policy on the caption. Therefore, once trained, our agent does not require language captions to perform well, and is not constrained by the potentially slow inference in a large vision-language model– it simply distills relevant knowledge from the model into its compact exploration policy. We include an architecture diagram in Figure S2.

---

### Author Response · Authors · 2022-08-08
**Call for Remaining Inquiries**

Dear reviewers,

Thank you again for your reviews -- please let us know if you have any remaining questions or concerns, so that we can address them before the deadline tomorrow. We hope that the new diagrams and supplementary figures add more clarity to the method/analysis. Alternatively, if you feel that your original concerns are addressed, we would appreciate updating your evaluation to reflect that.

Thank you so much,
The authors

---

### Meta-Review · Area_Chair_StXF · 2022-08-31

**Recommendation:** Accept
**Confidence:** Certain

**Metareview:**

The proposed work aims to address a concern in novelty-based RL exploration of weak representations for guiding exploration.  They leverage image captioning to provide a semantic categorization of states for defining novelty.  The reviewers were happy with the conceptual contributions as well as the experimental results on appropriate domains.

The "missing piece" is a better theoretical understanding of why this approach is good.  Specifically,  what about language is helpful and how might such properties be guaranteed in future trained captioning models?

**Award:**

No

---

### Decision · Program_Chairs · 2022-09-14

Accept